# New Form, New Material and Color Scheme, the Exposed Concrete Phenomenon—The Centennial Hall in Wrocław

**Jerzy Ilkosz [1], Ryszard Wójtowicz [2] and Jadwiga Urbanik [3],***

[1] Wroclaw Museum of Architecture, 50-156 Wrocław, Poland; ilkosz@ma.wroc.pl
[2] Drabik & Wójtowicz s.c.—Conservation and Restauration of Art, 51-113 Wrocław, Poland; zabytki@drabikwojtowicz.pl
[3] Faculty of Architecture, Wrocław University of Science and Technology, 50-370 Wrocław, Poland
* Correspondence: jadwiga.urbanik@pwr.edu.pl

**Abstract:** The aim of the article is to present the remarkable changes in architecture that took place in the 20th century. They can easily be called a revolution regarding the architectural form and the color scheme. Progress was being made through the development of reinforced concrete production methods. In the German Empire (*Deutsches Kaiserreich*), this material quickly found applications in more and more interesting solutions in architectural structures. In Wrocław (formerly Breslau), then located in the eastern German Empire, exceptional architectural works were realized before and after the First World War using new technology. In 1913, an unusual building was erected—the Centennial Hall, designed by Max Berg (inscribed on the UNESCO World Heritage List in 2006). Berg's work was inspired by the works of both Hans Poelzig and Bruno Taut. On the one hand, it was a delight with the new material (the Upper Silesian Tower at the exhibition in Poznań, designed by H. Poelzig) and, on the other hand, with the colorful architecture of light and glass by B. Taut (a glass pavilion at the Werkbund exhibition in Cologne). Max Berg left the concrete in an almost "pure" form, not hiding the texture of the formwork under the plaster layer. However, stratigraphic studies of paint coatings and archival inquiries reveal a new face of this building. The research was carried out as part of the CMP (Conservation Management Plan—prepared by the authors of the article, among others) grant from The Getty Foundation *Keeping It Modern* program. According to the source materials, the architect intended to leave the exposed concrete outside of the building, while the interior was to be decorated with painting, stained glass, and sculpture. The stratigraphic tests showed that the external walls were covered with a translucent yellowish color coating. Thus, the Centennial Hall shows a different face of reinforced concrete architecture.

**Keywords:** 20th century; Germany; German Empire; Silesia; Wrocław; Centennial Hall; concrete architecture; Max Berg; color scheme; CMP

## 1. Introduction

The history of twentieth-century architecture abounds in so many dramatic events, surprising works, original artists, and controversial ideas that getting to know these phenomena becomes a fascinating adventure.

Certainly, one of these surprising, original, and controversial projects is the Centennial Hall, designed by Max Berg and built in 1910–1913 as the main building of the Exhibition Grounds in Wrocław.

This is a unique example of the innovative use of reinforced concrete. Never before had there been the decision to show the texture of raw concrete with traces of the wooden formwork and combine it with a sophisticated color decoration—stained glass, sculptures, and paintings. Although the project has not been fully implemented, the novelty of the idea should be appreciated.

For this reason, in 2006, the Centennial Hall (Figure 1) was inscribed on the UNESCO World Heritage List as "one of the few examples of the architecture of the last century, due to its value as a monument to the human genius" (Ilkosz et al. 2016, pp. 135–36).

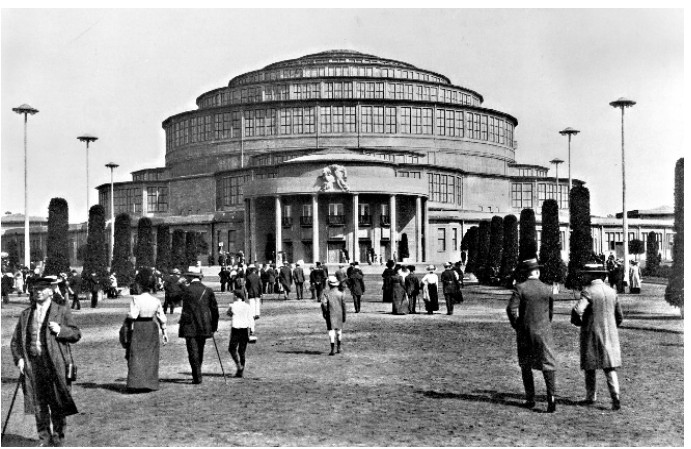 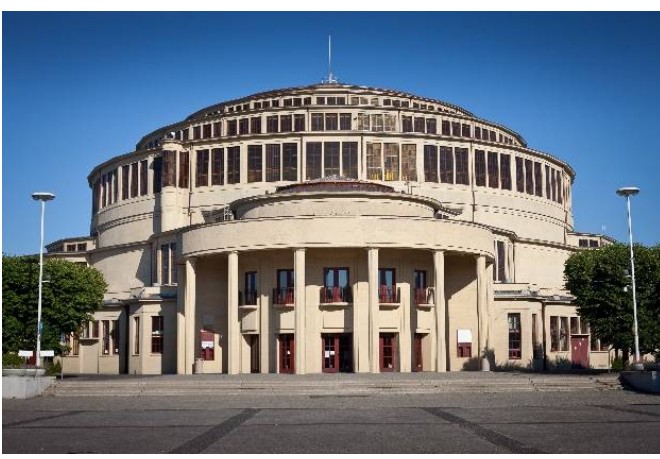

(**a**) (**b**)

**Figure 1.** Centennial Hall built in 1910–1913, designed by Max Berg: (**a**) during the Centennial Exhibition in 1913, *Dekorative Kunst*, 1912–1913, p. 537; (**b**) after renovation in 2009, contemporary view (photo M. Kulczyński).

So, it was not difficult to present the OUVs (Outstanding Universal Values) of this monument in the application. They indicate the authentic architectural form, the specific construction technique, and the materials used.

The most important publications for the development of this text are those on Centennial Hall. Although it is mentioned in many of the monographs on 20th-century architecture, it was not until the beginning of the 21st century that the history of Wrocław's Centennial Hall was more broadly elaborated on the basis of an analysis of archival resources and other sources, such as press publications (Ilkosz 2006; Ilkosz et al. 2016). Researchers usually emphasized only the novelty of the material and the construction.

The reasons for such a late publication of a monograph on the Centennial Hall include, above all, the geographical location of Wrocław on the periphery of what was then German Empire and the change of borders after World War II. As a result, not only the hall but also other works of Wrocław's modernism have only recently become better known to the world's scholars. The first descriptions of the hall appeared in the 1970s.

In 1971, Nikolaus Pevsner wrote: "The creator of the Centennial Hall in Breslau achieved with reinforced concrete between 1910 and 1912 what Behrens had achieved with steel structures. [ . . . ] The supports, the span and the curvature of the arches are characterized by an elegance that foreshadows the future achievements of Pier Luigi Nervi after the Second World War" (Pevsner 1971, pp. 162, 200).

Reyner Banham, comparing the hall structurally with the steel industry pavilions in the 1913 Leipzig exhibition and the glass pavilion in the 1914 Werkbund exhibition in Cologne, recalled: "No other work in Berg's long career, both as the city architect of Breslau and as an independent designer, can be compared with this enormous dome". In comparison with Centennial Hall, "Perret's contemporary work seems wooden and intellectually limited. Only here, nowhere else in Germany or elsewhere, has a building been built that withstands comparison with the Muthesius canon of nineteenth-century exhibition buildings in terms of scale, originality, and use of material" (Banham 1979, p. 82).

The authors writing about the hall pointed out the span of the dome, which was greater than the scale of all the earlier domes of historical buildings (the Pantheon, the Hagia Sophia, or St. Peter's Basilica) and gave numerical data and basic information about its construction, but few stressed the significance of the building for the development of

modern architecture, and few analyzed its relationship to the work of other architects of the time (Vischer and Hilberseimer 1928; Müller-Wulckow 1929; Schumacher 1935; Joedicke 1958; Roh 1958; Tintelnot 1959; Thiele 1963; Arnason 1970; Lampugnani 1999; Scheyer 1984; Hersel 1987). Werner Hofmann saw the hall as a futuristic work and compared it to Giacomo Balla's painting *Mercury Passing by the Sun* (Hofmann 1980, p. 423). Henry-Russell Hitchcock very accurately connected Berg's work with Friedrich von Thiersch's hall in Frankfurt on Main, built in 1907–1908. As a work of engineering, the hall in Wrocław refers to the great station halls of Stuttgart and Karlsruhe (Hitchcock 1987).

In the 1970s, the Centennial Hall and the architecture of Wrocław in the first half of the last century were discussed in more detail in monographs by Ernest Niemczyk. The author rightly noted that Berg's intention was to design "from the inside, for the inside" (Niemczyk 1972, 1978). He drew attention to the genesis of the construction of the hall when both Semper's and Wagner's ideas of the unity of arts were alive as well as to the combination of "classical archetypes (the dome and portico) with the Gothic ribbed construction", concluding that the designer of the building was close to Platonic ideas "about the objective beauty of regular geometric forms" (Niemczyk 1997a, 1997b).

Ewa Chojecka wrote from the perspective of the 1990s: "[ . . . ] it is difficult to agree with the opinions of Niemczyk, who interprets Centennial Hall as an expression of a utopian architectural idea oriented towards the values of socialism, democracy, and revolution, as these concepts were understood in the 1970s" (Chojecka 1997, p. 116).

The Centennial Hall, as a work of engineering art, is also very interestingly presented against the background of other buildings in the book by Erwin Heinle and Jörg Schlaich, which discusses the history of the dome in the architecture of different periods and cultures (Heinle and Schlaich 1996). They emphasized, as Ludwig Hilberseimer did even before the war, the exploratory nature of the construction and the use of new materials. The dome of the Roman Pantheon held primacy of size until the Centennial Hall, but fifteen years later, the Leipzig Trade Fair Hall, designed by Hubert Ritter and Franz Dischinger, received a reinforced concrete dome that was even lighter and had a longer span (Heinle and Schlaich 1996; Vischer and Hilberseimer 1928).

The historical background of the period and the local social relations interested the scholar of Expressionism in architecture, Wolfgang Pehnt, who, in his classic monograph *Die Architektur des Expressionismus*, presented Berg's work as "corresponding to the Wilhelminian need for representation" (Pehnt 1978, p. 366). Pehnt was of the opinion that the architect chose reinforced concrete as the construction material because it guaranteed the durability of the building, which will be "even after centuries a witness to the culture of our time" (Berg and Trauer 1913, p. 164). The hall is a *monumentum aere perennius* (Pehnt 1973, p. 69), for the essence of the hall is to show an unmasked construction, which, as in the Gothic, had its artistic expression, as Wilhelm Worringer put it in *Formprobleme der Gotik* (Pehnt 1973). In the book published again in 1998, Pehnt slightly expanded his thoughts on the Wrocław hall. Adopting the term "Zyklopenstil", after Karl Scheffler, for German architecture around 1910 as a manifestation of Wilhelmine monumentalism with its features of primitivism and decadence, he treated the Centennial Hall as a link between the Cyclopic buildings and the utopian designs of the Expressionists and their architectural visions (Pehnt 1973). Matthias Schirren has pointed out the influence of Centennial Hall and the buildings associated with the Centennial Exhibition on the work of Hans Poelzig and his interest in theater buildings (Schirren 1989).

Contemporary American architectural scholar Kathleen James likened the hall's importance to architecture to the importance of Kandinsky's paintings to modern painting. She wrote that "Berg combined abstract forms with an unparalleled feat of engineering, and the reinforced concrete ribs of the vault, the great arches, and the details create a similar effect to the abstract shapes in Kandinsky's paintings. The expression of the interior, achieved by rhythmically forming the exposed concrete structure, foreshadowed the dynamism in Erich Mendelsohn's work, already evident in his Einstein Tower in Potsdam" (James 1997, pp. 19–21; James 1999; James-Chakraborty 2000).

This text complements the knowledge about the Centennial Hall (as designed and implemented), especially when it comes to the combination of multicolored decorations with the rawness of concrete.

The presented knowledge is the result of extensive historical studies based on archival queries (the State Archive in Wrocław, the Wrocław Construction Archive, a branch of the Museum of Architecture in Wrocław, the Institut für Regionalenentwicklung und Strukturplanung in Erkner near Berlin, the Deutsches Museum in Munich, and the Archives in Berlin), iconographic research (documents, texts in the daily press, magazines, and compact publications), and literature studies of contemporary publications. Their results contradict the view, which existed until recently, that architects of that time were fascinated by raw concrete and its natural colors, which resulted from the ingredients used.

The most interesting information is the result of the concrete research and stratigraphic studies of paintings carried out during the restoration works of the building in 2009–2015.

## 2. The Beginnings of Reinforced Concrete in Architecture

When analyzing the first reinforced concrete structures, special attention should be paid to the aesthetic role of the new material. Concrete has been known since ancient times, but it was not until the second half of the 19th century that all its properties were known and used in combination with iron. Since Joseph Monier filed his patent in 1867 demonstrating the versatility of the new material, architects have been given entirely new possibilities for both construction and aesthetics.

The new material first appears in the realization of mills, granaries, silos, tanks, factories, garages, and bridges, where function did not require detail and decoration. The villa François Hennebique built for himself in the 1890s in Bourg-la-Reine shows almost all the structural possibilities of using reinforced concrete. The architectural form, however, still conforms to the style of the period. In 1900, in Danton Street in Paris, he realized a residential block with a reinforced concrete structure, the external appearance of which did not differ from a normal brick house (Biegański 1972).

The first reinforced concrete building to take full advantage of the new material is considered to be the Saint-Jean de Montmartre church in Paris, designed by Anatole de Baudot in 1894–1904. The architect used reinforced concrete to design the skeleton of the building, which was enclosed on the outside by thin walls, thus giving it the form of a traditional building. However, the interior shows modern construction and raw concrete (Biegański 1972).

It was not until the twentieth century that the aesthetic qualities of reinforced concrete were appreciated and the first realizations appeared, revealing this material on façades. Initially, and still tentatively, it was covered either with plaster or relief, or it was given the form of another material, using stone techniques (De Jonge 2021).

The use of reinforced concrete coincided with the desire of many prominent early-twentieth-century architects for a simple form that conformed to the tenets of functionalism.

Developments using reinforced concrete for construction appeared throughout Europe. The first residential building in Europe with a reinforced concrete structure was the house in Paris on Franklin Street, designed in 1902–1904 by August Perret (Giedion 1968; Biegański 1972). As never before, the facade exposed the structure, which was betrayed by the proportions between the full wall and the window area. However, the structural elements visible from the outside were covered by the architect with terracotta tiles decorated with a plant motif (Koren 2019). This is the first example of the use of reinforced concrete as a means of architectural expression. This solution opened a new path that the entire architectural world soon entered.

The *Theatre des Champs Elysées* in Paris, designed by the Perret brothers (Auguste and Gustav) in 1910–1913, was made entirely of a reinforced concrete construction, which made it possible to realize the largest auditorium at the time. Its external form, however, with flat-sculpted decoration and stone cladding, does not yet show the severity and simplicity of the material (Biegański 1972).

It can be assumed that from the end of the 19th century (1894), reinforced concrete—an invention of the French technical world—began to be used throughout the world.

In the German Empire, reinforced concrete was used very quickly, before the First World War, both in the construction of buildings and in increasingly interesting architectural solutions.

The Leipzig railway station building (1906–1915), designed by William Lossow and Max Hans Kühne, and the Stuttgart railway station building (1911–1928), designed by Paul Bonatz and Fridrich Eugen Scholer, are mainly mentioned in the literature (Pehnt 2006). In both cases, the modern construction is not visible from the outside.

The first church with a reinforced concrete structure was the garrison church in Ulm (1908–1910), designed by Theodor Fischer. Although the structure was shown on the façades of the building, its form has the characteristics of traditional brick architecture (Wertz 1970).

An interesting construction is the trade fair hall in Munich, designed between 1908 and 1912 by Richard Schachner. The hall was 98 m long and, although it was covered by a traditional gable roof, presented an extremely simple modern form. The gable walls, fully glazed, were devoid of any architectural detail, and the side walls had a series of simple windows only in the upper part. This building fully presaged the domination of function over form in the architecture of the later decades of the 20th century (Schachner 1910).

In Breslau, even before the First World War, the use of a new material appeared. Between 1905 and 1913, in just eight years, several buildings were designed that played an important role in the history of European architecture.

The most important of these was the Centennial Hall. However, the hall was preceded by other buildings of the city. The Breslau theater (*Metropol*, later *Schauspielhaus*, and today *Teatr Polski*), built in 1905–1908 and designed by Berlin architect Walter Hentschel, was the first in Europe to have a stage with a reinforced concrete structure (Figure 2). However, its external walls were still decorated in the spirit of Art Nouveau (Jagiełło-Kołaczyk 2011).

The Market Halls (Figure 3), dating from 1907 to 1908, were among the first in Europe to use reinforced concrete structures of parabolic arches with a span of 19 m. Richard Plüdemann and Heinrich Küster (Gryglewska 2008) left the concrete inside in an almost "pure" form, not hiding the texture under a layer of plaster. Only a modest polychrome with floral motifs was made on the underside of the arches. The buildings still had the traditional external "costume". The facades of the halls represented historicist architecture, while the interiors foreshadowed the architecture of the famous Centennial Hall by Max Berg, with their concrete parabolic arches reminiscent of the dome and monumental arches of the Centennial Hall.

Almost at the same time, another outstanding architect working in Breslau, director of the Royal (later State) Academy of Arts and Crafts (*Königliche/Staadliche Akademie für Kunst und Kunstgewerbe*), Hans Poelzig, succumbed to his fascination with reinforced concrete. This architect first used concrete as a building material both for the construction of buildings and for the interior design in 1911–1912 in the famous office building in *Junkernstrasse* (today's Ofiar Oświęcimskich Street), where he tried to form the modern material in reference to the spirit of the wooden fachwerk tenements with their characteristic overhanging of successive stories and decorative shaping of the ends of beams (Schirren 2000; Janas-Fürnwein 2000a, 2000b; Ilkosz and Krzywka 1997).

Hans Poelzig's office building (Figure 4) was already modern in both interior and façade design. The office building foreshadowed the department stores and metropolitan architecture designed in the 1920s by, among others, Erich Mendelsohn. It was also the first reception in Europe of American architecture of the so-called Chicago school, which developed in the United States at the turn of the 19th and 20th centuries. Poelzig's building is compared to an icon of 20th century architecture—Walter Gropius's Fagus factory in Alfeld an der Leine.

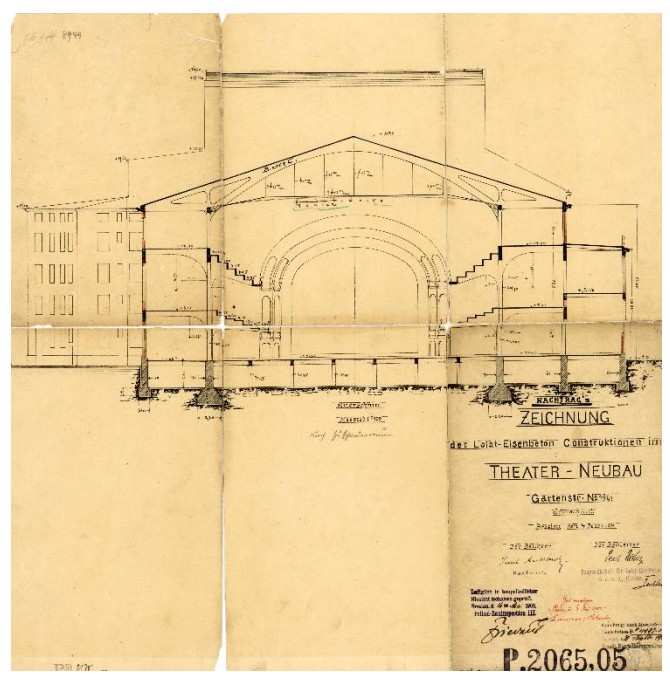 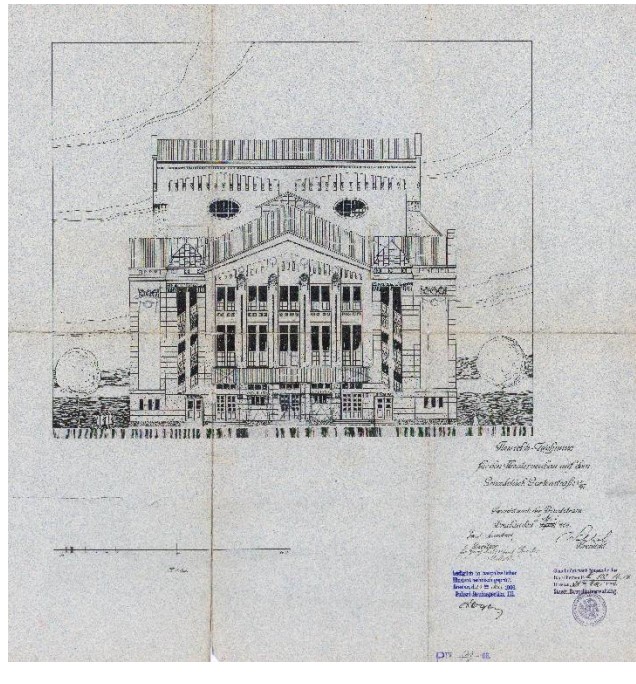

(**a**)                              (**b**)

**Figure 2.** The theater (*Metropol*, later *Schauspielhaus*, and today *Teatr Polski*), built in 1905–1908 and designed by Berlin architect Walter Hentschel: (**a**) cross-section and (**b**) front elevation (Wrocław Construction Archive, a branch of the Museum of Architecture in Wrocław).

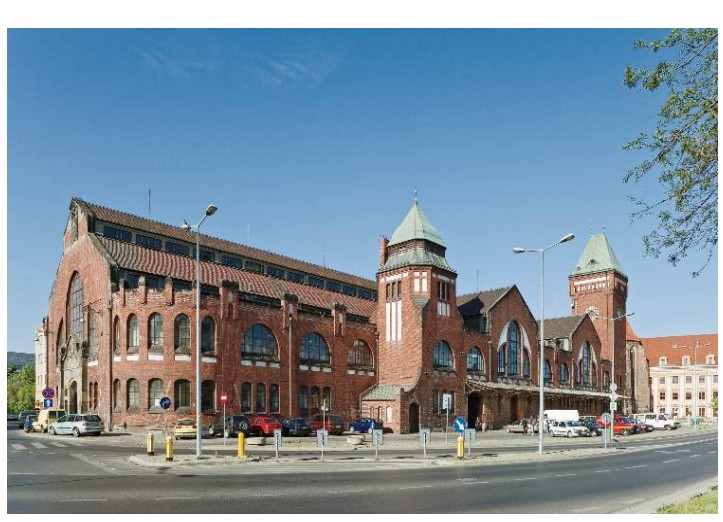 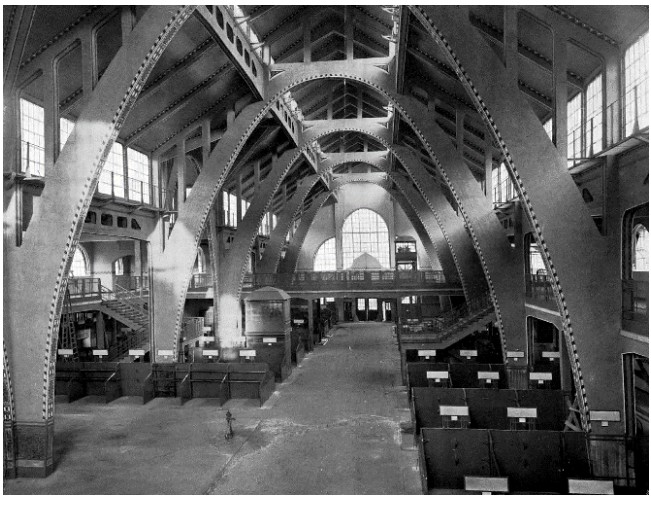

(**a**)                              (**b**)

**Figure 3.** The Market Hall, built in 1907–1908 and designed by Richard Plüdemann and Heinrich Küster: (**a**) contemporary view (photo S. Klimek) and (**b**) interior view (Küster 1909, p. 97).

In 1912, Hans Poelzig became involved in the design process of the Breslau Exhibition Grounds, for which he created the Pavilion of the Four Domes (Figure 5) and the pergola (Figure 6). His work, like the hall, shows raw concrete from the outside, the only difference being that some of the architectural elements were developed using stone techniques.

What all these buildings have in common with the Centennial Hall is the pioneering use of reinforced concrete as a new and modern building material, as well as the innovative shaping of the architectural form, which includes a break with decorativeness, designing according to the principles of "organic architecture" or "from the inside out", and empha-

sizing the functionality and construction of the building. All these projects were created in the spirit of the social and artistic reformatory currents of the early 20th century. They heralded revolutionary changes in 20th-century architecture, which became apparent after World War I, especially in the 1920s in the so-called modern construction *Neues Bauen*, also referred to as the International Style or expressionism.

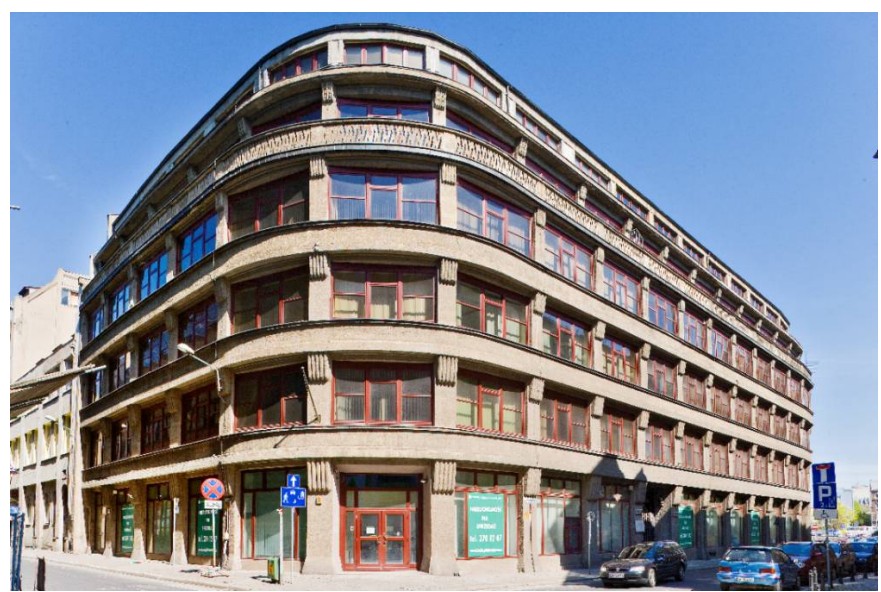

**Figure 4.** Office building in *Junkernstrasse* (today's Ofiar Oswięcimskich Street), built in 1911–1912 and designed by Hans Poelzig, contemporary view (photo S. Klimek).

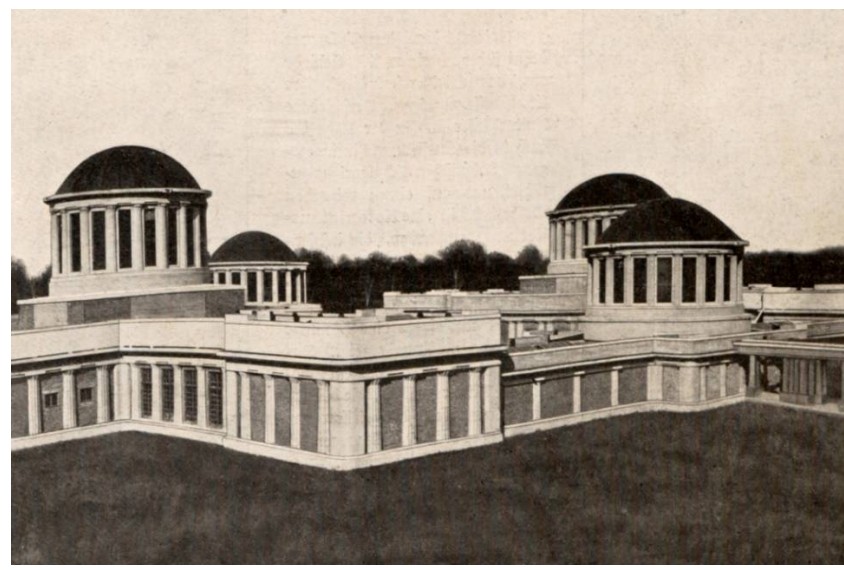

**Figure 5.** Pavilion of the Four Domes, built in 1912–1913 and designed by Hans Poelzig, Breslau Exhibition Grounds (Buchwald 1913, p. 17).

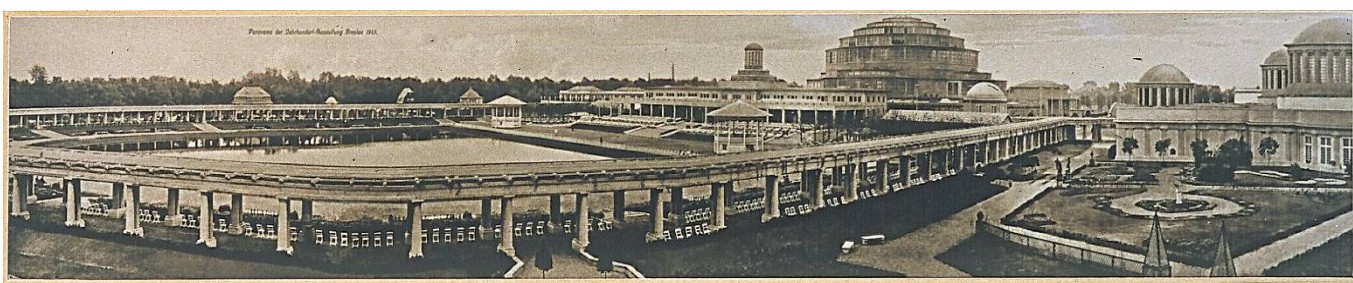

**Figure 6.** Pergola built in 1912–1913 and designed by Hans Poelzig, Breslau Exhibition Grounds (Museum of Architecture in Wrocław).

However, in none of the discussed buildings did the authors foresee a combination of the raw material, which was concrete, with multicolored decorations. For the first time, Max Berg proposed such a solution, unusual for those times, in the crematorium and the Centennial Hall designed for Wrocław.

An exceptional work of Max Berg's was a design of a crematorium (not realized) for the city of Wrocław from 1912 to 1918 (Ilkosz 1996). The conception of this building came directly from Berg's reflections on the Centennial Hall. It was planned as a monumental reinforced concrete building (Figure 7). The crematorium and the Centennial Hall are excellent examples of an attempt to connect and integrate visual arts with architecture and also an attempt to introduce color into reinforced concrete architecture. The design of crematorium was created by Max Berg together with an outstanding Viennese painter Oskar Kokoschka, who was supposed to design a fresco decorating the interior.

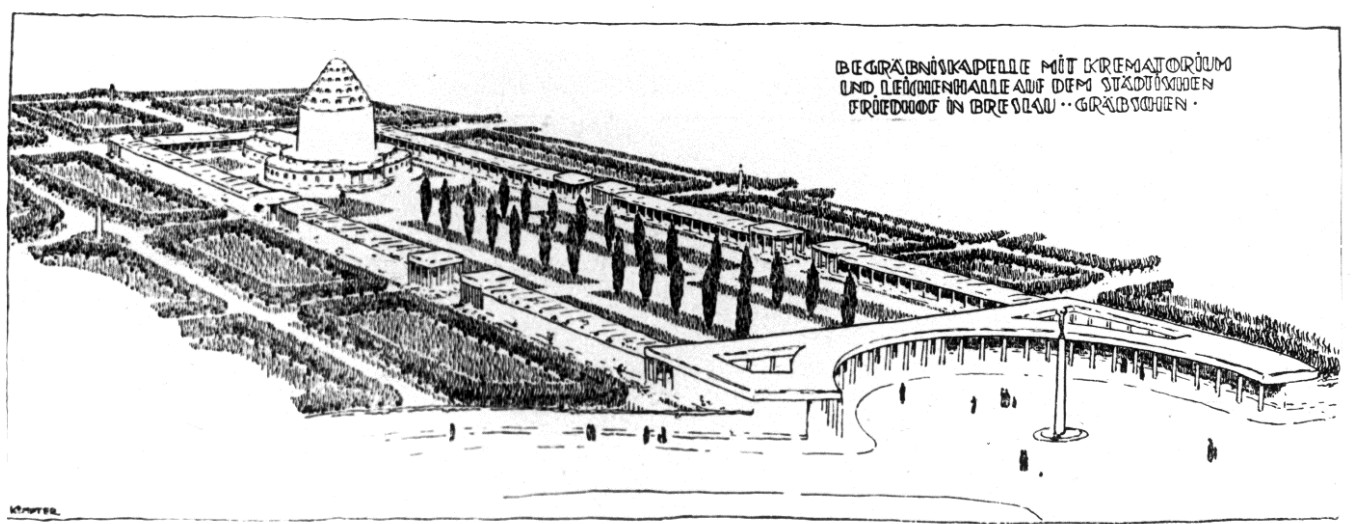

**Figure 7.** Funeral chapel with crematorium and mortuary designed by Max Berg, in the municipal cemetery in Breslau Gräbschen (Deutschland Städtebau Breslau, Berlin, 1921, p. 41).

## 3. Centennial Hall

### 3.1. Form and Ideological Content

The creation of the Centennial Hall was an exceptional event in the architectural world. The Centennial Hall was designed in 1910 by Max Berg in cooperation with Richard Konwiarz and Günther Trauer. Its construction was connected with the organization of the Centennial Exhibition in 1913, which was to commemorate the 100th anniversary of the victory over Napoleon. At the same time, the exhibition was to initiate the creation of an exhibition area in Wrocław, following the example of Leipzig and Poznań. Construction works started in 1911 and were completed in May 1913 by the following companies: *Dyckerhoff & Widmann* from Dresden and *Lolat Eisenbeton Breslau A.G.* from Wrocław. Construction

supervision on behalf of the city was entrusted to Günther Trauer and Paul Schreiber. Ernst Matthes was responsible for the construction on behalf of the city, and engineer Meyer acted on behalf of *Dyckerhoff & Widmann*. The Centennial Exhibition opened on 20 May 1913 (Ilkosz 2006; Ilkosz et al. 2016). The Centennial Hall was the world's first public building made in reinforced concrete on such a large scale.

The hall was founded on a symmetrical four-leaf plan (tetrakonchos), formed by an inner circle and four apses opening into it on a semicircular plan. The inner circle forms the basis of a dome with a diameter of 65 m. The interior of the hall is 42 m high, of which 19 m falls on the base and 23 m on the dome proper. The whole is surrounded by a walkway repeating the quadrilateral outline. Thanks to the glazing of the walls and the openwork construction inside, the building impresses with its lightness of form (Figure 8).

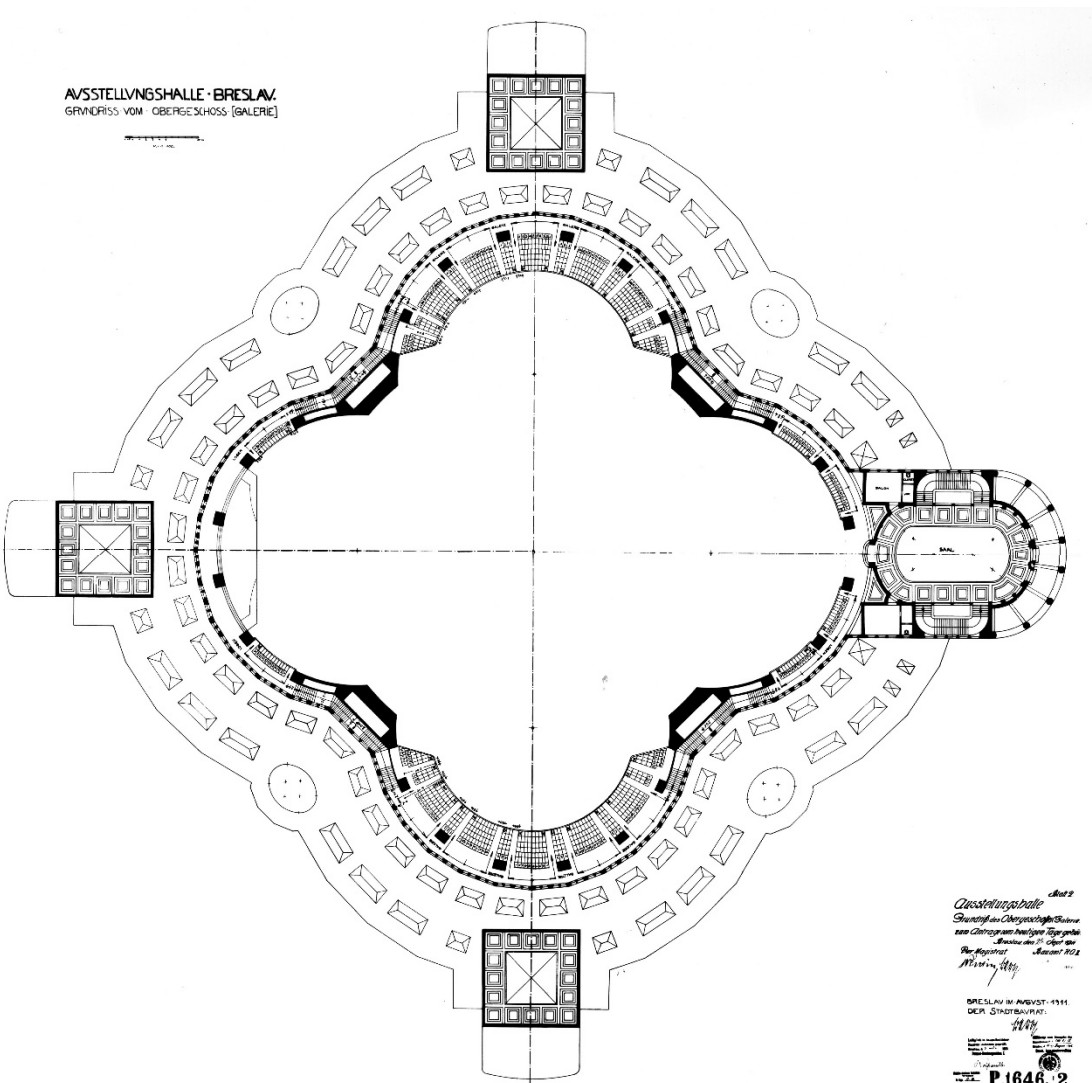

**Figure 8.** Centennial Hall, plan of second floor and galleries, developed in August 1911 and approved by the Municipality of Wrocław on 25 September 1911 (Wrocław Construction Archive, a branch of the Museum of Architecture in Wrocław).

Like the crematorium, created in Berg's collaboration with Oskar Kokoschka (1912–1916), the hall was to be an ideal building dominating the landscape. The design of this building was an exemplification of the "crystal rock", symbolizing through its "creative forces" nature, the source of new life, and new art at the same time (Wolter 1955). The characteristic motif of a mountain draws attention to the fascination of Berg and other archi-

tects of his time, such as Fritz Schumacher, Bruno Taut, Wenzel Hablik, Erich Mendelsohn, Henry van de Velde, and Rudolf Steiner, an anthroposophist and architect, with Friedrich Nietzsche and his book *Thus Spoke Zarathustra* (*Also sprach Zarathustra*). In a letter to his younger collaborator, Paul Heim, Berg wrote: "Architecture should find a place similar to other arts in order not to become a mere craft. Although the language of architecture is different, so are the possibilities of spiritual influence, which do not lie in the realm of sensuality, but move the deepest feeling, a spiritual divine source that flows from cosmic forces, truth and love" (Šlapeta 1989, p. 28).

When designing the Centennial Hall, Max Berg referred to antiquity, looking for the prototype in the Pantheon. The form of the hall is also close to the central Byzantine buildings, especially the church of Hagia Sophia in Constantinople, with which it competes by the span of the dome. Analogies with St. Peter's Basilica in Rome are also evident, especially with the designs of Donato Bramante and Michelangelo. The plan of the hall also resembles in some elements the plan of the Renaissance church of Santa Maria della Consolazione in Todi or even seems to be an imitation of the early Christian San Lorenzo in Milan. This recourse to classical tradition was a search for a new style corresponding to the present, which would be a contrast to the passing 19th century and the ephemeral Art Nouveau, and at the same time could be the leaven for the emerging avant-garde. The search for simplicity and the truth of the material was the slogan of the era. It was called for by proponents of neo-Biedermeier as well as architects breaking with historicism. Berg used the tradition of antiquity and outstanding achievements of later eras to seek inspiration for constructing the entire building according to the principles of proportion and size of solids and was guided by the utopian idea of creating an absolute form. Reminiscences of Egyptian and Mexican architecture can also be seen in the Berg's designs.

On the one hand, when designing the Centennial Hall, Berg was fascinated with noble proportions; the golden section, symmetry, simple geometrical figures, and numerology are indicative of his pursuit of permanent principles in both the aesthetic and the ethical sense. Berg reaches to Plato's idealist philosophy. The Centennial Hall's arrangement as a domed building laid out on a quatrefoil referred to the architect's platonic view of symmetry as a cosmic, eternal, and sacred principle of nature. However, on the other hand, like Berlage's theories, the individualist and romantic themes in Berg's views on art reflect his fascination with the Gothic. To Berg, who had studied under Carl Schäfer, a leading advocate of the Neo-Gothic in 19th-century Germany, Gothic means "paradise lost"—the last grand style. Berg appreciated Gothic architecture for its approach to space. The search for the order and the harmony of the "grand style" to inspire the new architecture informed the pursuits of Max Berg and his contemporaries (Walter Gropius, Brunon Taut, and Erich Mendelsohn) and led them to believe in utopian ideas. Wolfgang Pehnt has said that they identified this "paradise lost" with India or the Gothic and expected "the coming of a new epoch, rebirth, time of spirit, faith, love, and the community of nations" (Pehnt 1973, p. 29).

Berg's design for the Centennial Hall reflects his search for an ideal monumental form. The influence of the architectural monuments of the Wilhelmian period cannot be disregarded. According to the contemporary critic Robert Breuer, the Centennial Hall "has been built like the pyramids: it is larger than required by its function and it serves the sublime idea as a monument to the German nation rather than honours the bygone past. Thus, like the Gothic cathedral in the Middle Ages, the Centennial Hall is a symbol of the new times" (Breuer 1912–1913, p. 519). Around 1910, monumentality became the order of the day. Peter Behrens asserts: "monumental art is the supreme and most accurate expression of the culture of our time" (Breuer 1912–1913, p. 72). Designs by Wilhelm Kreis, Otto Rieth, or Bruno Schmitz conform to the Cyclopean style. This idiom also informs Franz Schwechten's Imperial Palace in Poznań or his Kaiser-Wilhelm-Gedächtniskirche in Berlin and above all the memorials of the Wilhelmian period. "The memorial fever peaked during the competition announced in 1909–1910 for a national memorial conceived as a monumental thanksgiving symbol" (Pehnt 1973, p. 75). Among the 379 artists participating in the competition for the national memorial at Elisenhöhe by Bingen were the future

leading figures of the Modernist avant-garde: Ludwig Mies van der Rohe, Walter Gropius, Adolf Meyer, and Hans Poelzig (Pehnt 1973, p. 75).

The comparison of three heroic monuments commemorating the Napoleonic Wars—Leo von Klenze's Befreiungshalle (Liberation Hall) at Kelheim, Bruno Schmitz's Völkerschlachtdenkmal (Battle of the Nations Memorial), and Max Berg's Jahrhunderthalle (Centennial Hall) in Wrocław—clearly shows the different political motivations underlying their foundation and reception. Von Klenze's rotunda at Kelheim, elevated on a rock overlooking the Danube, was erected in 1842–1863. It symbolized the independent statehood of Bavaria and was King Ludwig I's gift to the nation, addressed primarily to the patriotically minded bourgeoisie and intelligentsia. Half a century later, the contemporaries perceive von Klentze's Pantheon and the Centennial Hall as underlain by similar ideas and representing the events of 1913 as the nation's spiritual victory and the triumph of its democratic aspirations. Bruno Schmitz's Battle of the Nations Memorial expressed a different perspective, glorifying the German nationalism propagated by the imperial court. These two approaches addressed the political situation of the German Empire (*Deutsches Kaiserreich*) (and of Wrocław) and the interpretation of Germany's later history and its unification under the leadership of Prussia. According to the conservatives, the leading role in the victory over Napoleon was played by the King of Prussia; the Liberals and Social Democrats glorified "the spirit of the nation" as the principal factor. The liberal City Council of Wrocław, dominated by the Social Democrats who held almost half of the seats following the 1907 election, interpreted Friedrich Wilhelm III's address of 1813 as his partial cession of the matters of state into the hands of the nation. To them, Friedrich Wilhelm III had emphasized the role of the nation at a crucial moment of history and fostered its emancipation.

### 3.2. Color Scheme

Centennial Hall is usually seen through the prism of functionalism, innovative construction, and the manifestation of the aesthetics of the material—reinforced concrete. Meanwhile, the source materials show that the architect intended to decorate the interior using polychrome, stained glass, and sculpture rather than architectural detail. His concept was not realized due to lack of funds. The city council, approving the construction of the hall, hoped to receive financial support from the Prussian government or even the emperor. However, this support was never obtained. The minimization of the artistic program of the hall was also influenced by the short time for the realization of this investment. The architect's intentions to expand the aesthetic program are confirmed by surviving drawings, as well as his comments in articles and correspondence. In a letter to Professor Sesselberg (18 February 1941), he wrote: "My intention was to give the whole building, by means of painting and sculpture, a symbolic expression for the spiritual sphere of man" (the architect's legacy at the Deutsches Museum in Munich, 050/001). In turn, writing to Paul Heim, he stated: "In designing the hall, I thought of placing the greatest emphasis on color by introducing stained glass and colorfully worked pendentives and the dome vault" (Šlapeta 1989, p. 28). Berg was probably inspired by the need, emphasized by the reformers, to enrich the aesthetics of gray tenement houses with appropriate colors. In 1901, Fritz Schumacher suggested that the color solutions of architecture should already be created during the design process. It remains an open question, however, what concept of decoration Berg chose for the Centennial Hall. The materials preserved in the artistic legacy of the architect, located in the *Deutsches Museum* in Munich, and concerning the crematorium designed together with Oskar Kokoschka in 1911–1916, may be helpful here (Ilkosz 1996).

In the years 1911–1912, Max Berg was connected with the circle of artists of the Berlin *Sturm*. It was then, probably through Herwarth Walden, that he met Oskar Kokoschka and perhaps came into contact with the architects later associated with the *Gläserne Kette* or with Paul Scheerbart, the visionary of "glass houses".

In April 1912, in a letter to Herwarth Walden, the publisher of the magazine *Der Sturm*, which promoted and animated new art in Berlin, Kokoschka mentioned a commission that was related to a Berg project (Spielmann 1992–1993). The painter did not specify it; perhaps he was already thinking about a crematorium. It seems very likely, however, that it was also about collaboration on the Centennial Hall. In August 1911, work began on the construction of the hall, and in April 1912, formwork was being prepared for the ribs of the dome (Ilkosz 2006). Berg could already think about the color scheme of the interior. In a letter to Alma Mahler from May 1914, Kokoschka wrote that he had received a letter from Berg in which the architect asked him to paint a 3 m × 3 m view of the Centennial Hall, which he wanted to present at an exhibition in San Francisco (Ilkosz 2006).

In September 1914, in a letter addressed to Kurt Wolf, a publisher of expressionist writers and poets, Oskar Kokoschka recalled: "Before the outbreak of the war I received a commission from Berg of Breslau to prepare sketches for a monumental painting for the newly built crematorium, he also made me hopeful of decorating the *Festhalle* there" (Kokoschka 1984). If the work had been done by Kokoschka, it would have been painting and stained glass in an expressionist style. The stained glass would probably have replaced the windowpanes, and in the dome, the polychrome would have decorated the pendentives of the great arches and the ceilings of the individual steps of the dome. This is also indicated by Berg's handwritten drawing in the margin of the manuscript on the acoustics of the Centennial Hall, as well as by some of Kokoschka's surviving sketches (Max Berg legacy at the Deutsches Museum in Munich). It should be emphasized that the structural elements of the hall were to be kept in raw concrete with traces of shuttering. The planned decorations were limited to the windows and the elements, which were covered with material supporting the acoustics of the building (Figures 9–11).

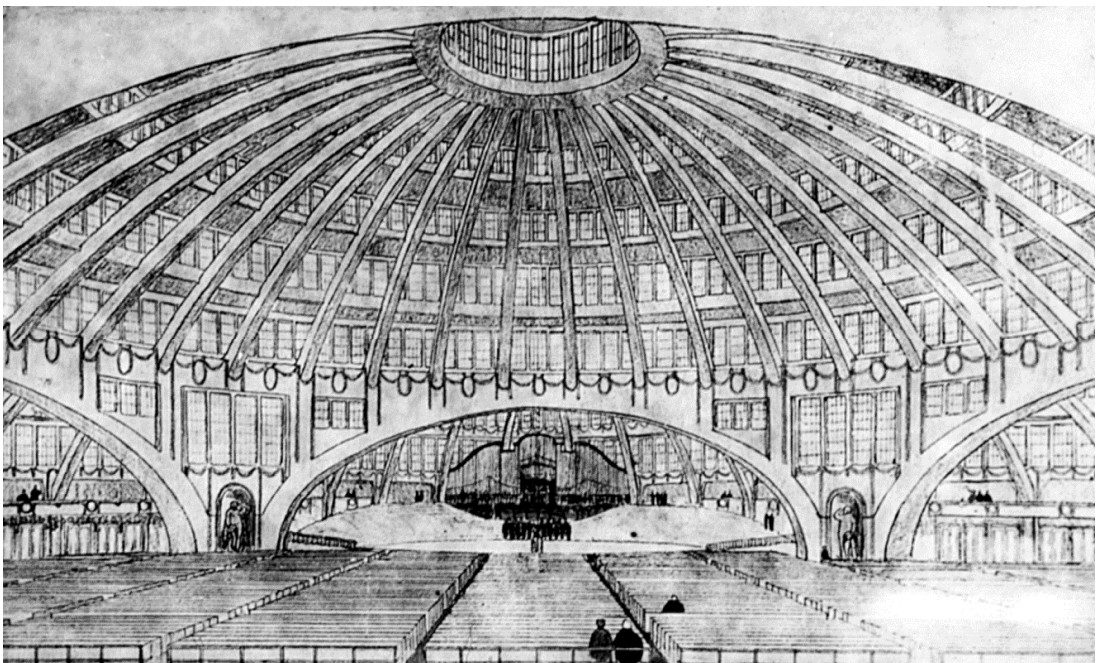

**Figure 9.** Design of the Centennial Hall interior by Max Berg, around 1910; a reproduction from a slide used by Berg for his lectures at the beginning of 1911 (National Museum in Wrocław).

In May 1914, Oskar Kokoschka received from Max Berg the finished designs of the crematorium, made back in 1912, and at the same time an invitation from the city authorities to come to Breslau. In a letter to Alma Mahler, he wrote: "Today I received the plans for the crematorium and a letter from Berg in the form of an official invitation. He is a true lover of art, who does not wait for the effect, but helps to make it a reality" (Kokoschka 1984). Around 23–24 May 1914, Kokoschka was in Breslau to conduct negotiations on the

frescoes for the crematorium (Kokoschka 1984; Strobl and Weidinger 1994). The surviving drawings show that he made sketches of the frescoes in two stages: May–July and fall 1914 (Spielmann 1992–1993). The three cartons from November 1914, now stored in Dresden, and some drawings from the collection in Vevey, also indicate that Kokoschka was not only interested in painting but also made architectural drawings (Figure 12). In November 1916, in a letter to his parents, he mentioned that during a meeting with Berg and his wife in Berlin, he showed his architectural designs for the crematorium. Berg praised him for his courage and good idea, then took these sketches to make technical drawings based on them in his office in Breslau (Kokoschka 1984).

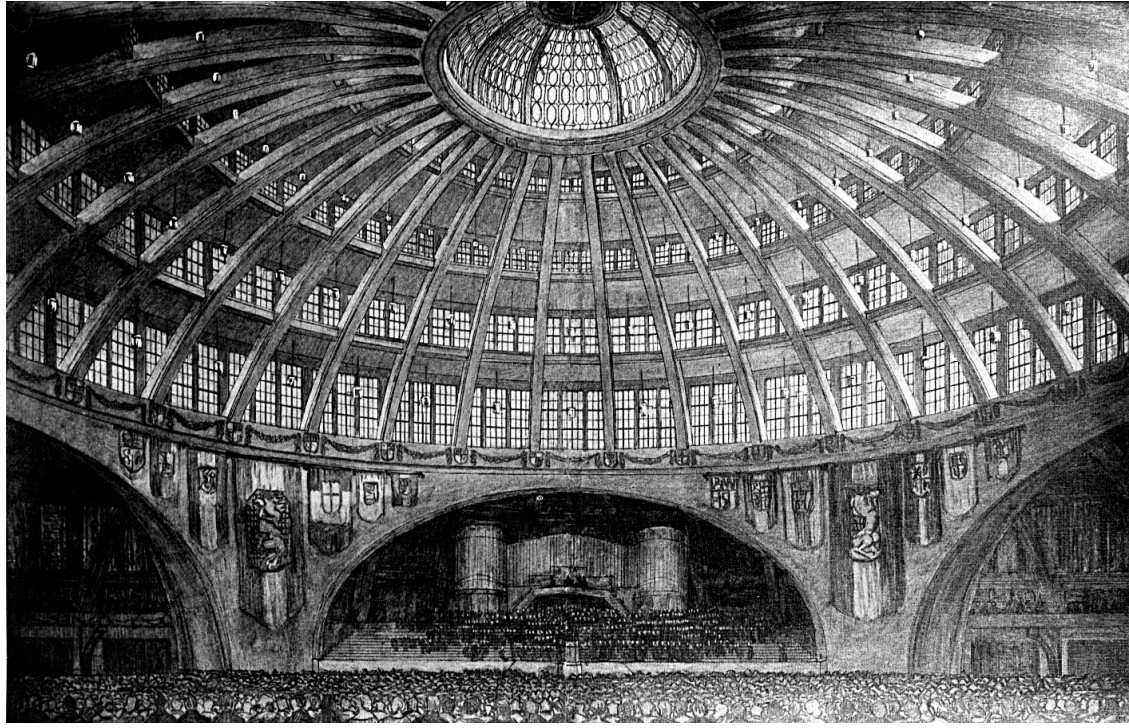

**Figure 10.** Design of the Centennial Hall interior by Max Berg (Institut für Regionalentwicklung und Strukturplanung in Erkner near Berlin).

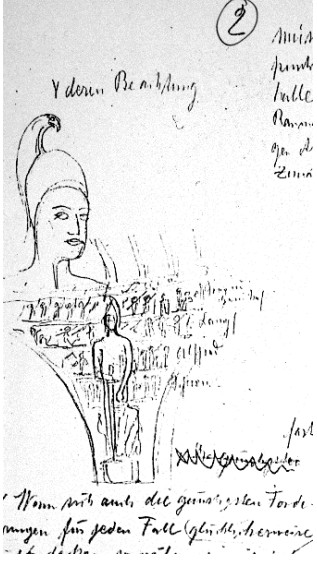

**Figure 11.** A sketch for the interior decoration of the Centennial Hall (Deutsches Museum in Munich).

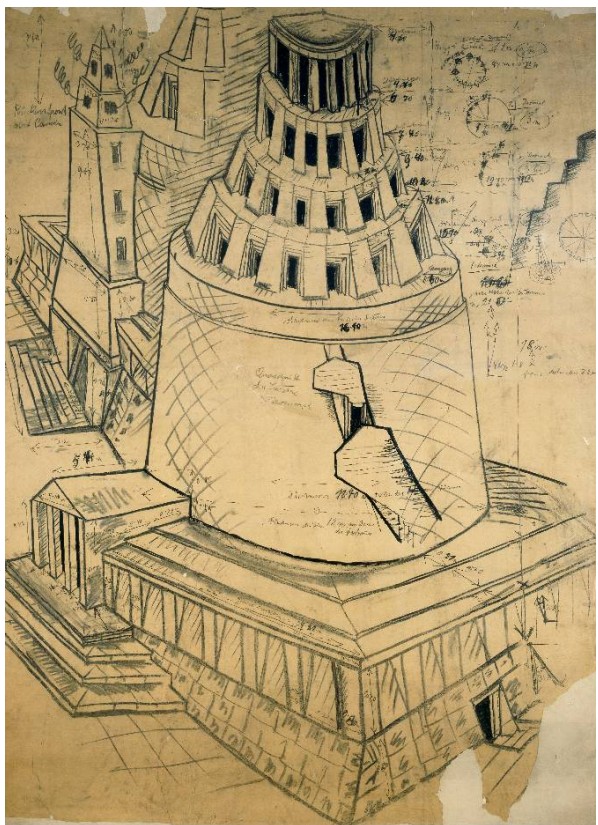

**Figure 12.** The crematorium designed by Oskar Kokoschka from around November 1914, based on Berg's designs (Staatlichen Kunstsammlungen Dresden).

Assuming that the drawing sketches from the *Deutsches Museum* concern the crematorium, it can be assumed that Berg initially presented Kokoschka with three preliminary design versions. They were probably intended to help the painter find a suitable artistic and iconographic form for the crematorium frescoes. It was probably for this reason that he travelled to Italy with Alma Mahler in 1913 to study Giotto's wall paintings. The sketches created during this trip testify to his search for artistic inspiration.

Berg's design marked the spot intended for Oskar Kokoschka's fresco. It was a wall about 13 m high and nearly 43 m long. Like the architectural design, the sketches for the frescoes developed in several phases, varying in composition and content. In November 1914, Kokoschka created further architectural drawings for the crematorium (now located in Dresden), which, drawn in an expressionistic manner with some exaggerations and deformations, show Berg's revised conception. The joint designs by Oskar Kokoschka and Max Berg illustrate the clearly designed type of decoration (Ilkosz 1996).

Berg probably gave Kokoschka a version of the crematorium that corresponds with the design from July 1914, which can be found in the Wrocław Museum of Architecture (a branch of the Wrocław City Building Archive) (Figure 13). It shows the building set on a square or rectangular plan (the projection has not survived) with a centrally located tower chapel, surrounded by a courtyard and columbarium cloisters open to the outside. The whole is set on a high and massive 1.60 m high pedestal. From the front elevation, facing west, the main alley of the newly established cemetery led towards today's Grabiszyńska Street. The entrance to the funeral chapel was preceded by a four-column portico, to which stairs led. The elevation was divided vertically by pilaster strips or half-columns in the great order. The chapel had the shape of a simple cylinder or cuboid, which narrowed in five steps from the middle of its height. A roof lantern rose above the narrow part. According to the project's dimensions, it was to be a monumental building with the diameter of funeral

chapel being about 14 m and the base width being about 29 m; the height of the entire building was to be 32.80 m (Figure 14a).

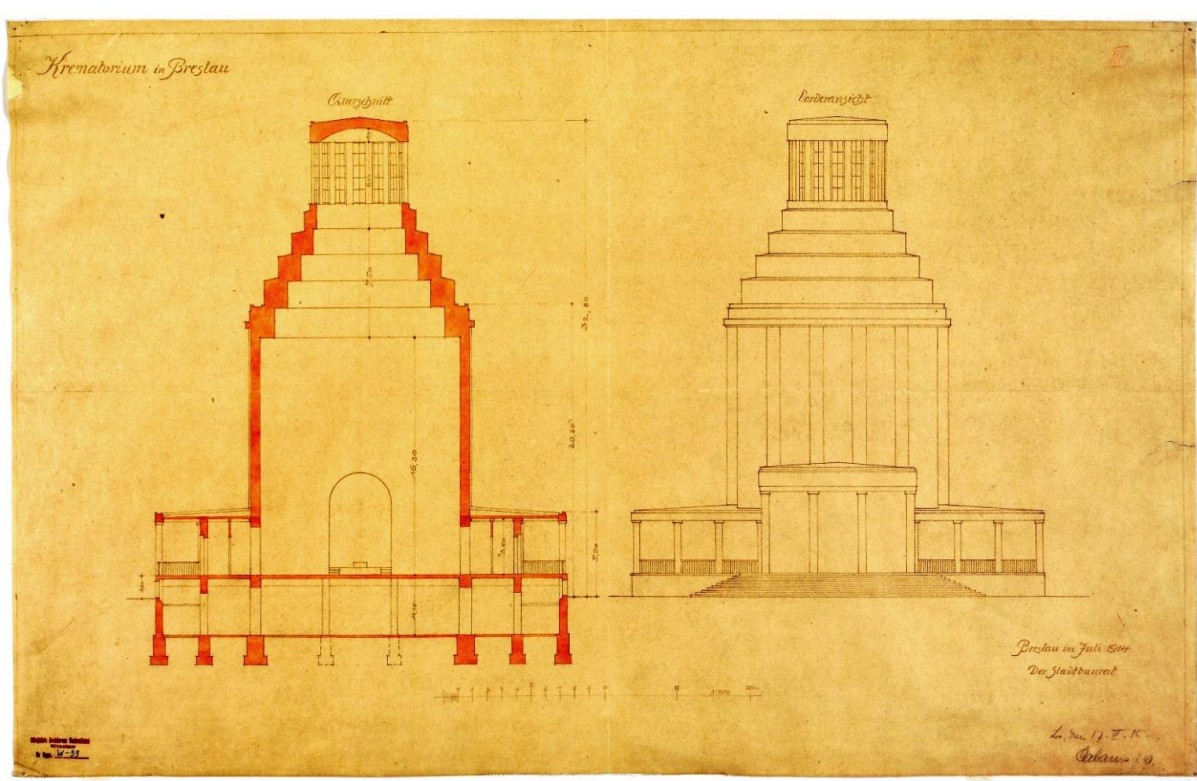

**Figure 13.** The crematorium design by Max Berg, cross section and front elevation, July 1914 (Wrocław Construction Archive, a branch of the Museum of Architecture in Wrocław).

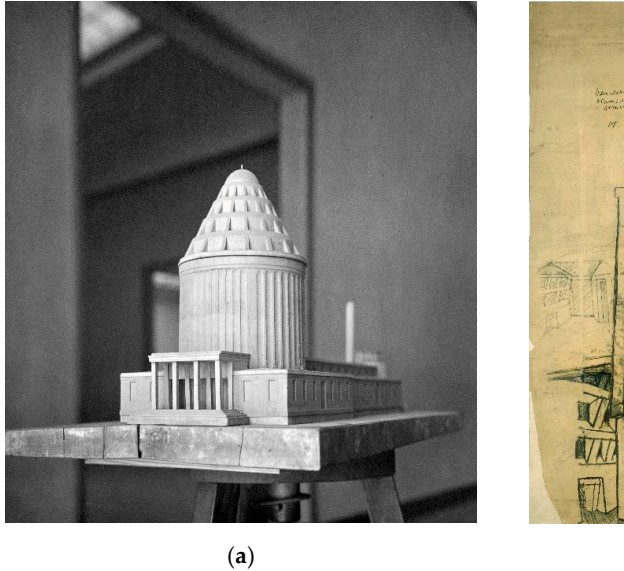

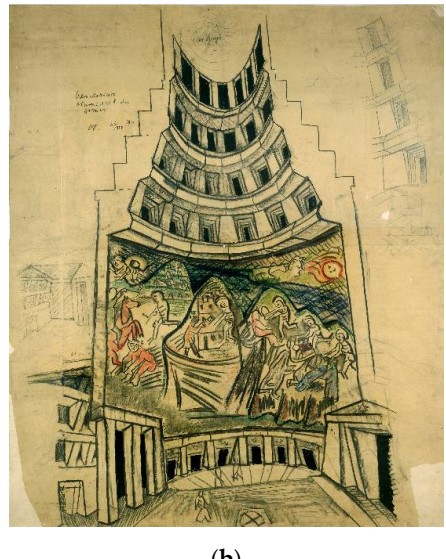

(**a**)                                                    (**b**)

**Figure 14.** The crematorium design by Max Berg: (**a**) model (Deutsches Museum in Munich); (**b**) painting decoration design by Oscar Kokoschka, 26 November 1914 (Staatlichen Kunstsammlungen Dresden, after http://www.7cudow.eu/pl/eksponaty/max-berg-oskar-kokoschka-niezrealizowany-projekt-krematorium-na-cmentarzu-grabiszynskim/, accessed on 27 November 2021).

Kokoschka's expressionistic sketch for an unrealized fresco for the wall of the crematorium refers to medieval representations of the triumph of death. As emphasized by

the monographers of Kokoschka's work, the frescoes were inspired by Giotto's paintings from the della Arena chapel in Padua. He must have been particularly influenced by the Last Judgment (Strobl and Weidinger 1994), as he transposes this theme in many sketches, replacing Christ in the mandorla with a symbol of human fate—Fortune on the globe. It was about changing the religious dimension to a more secular one. This was mainly due to the fact that the crematorium was also supposed to serve non-believers, and in view of the negative attitude of Catholics at the time to the cremation of the body, it was quite an important factor. Kokoschka made some sketches with the figure of Fortuna. He tried to show the omnipresence of death in all phases of human life, from birth to old age. The individual scenes correspond to human experience, joys and sorrows, love, and work. Death is surprising at every stage of life. Kokoschka accepted death as the complete end of human existence, with no hope of a resurrection.

The sketch shows saturated cool colors: greens and blues, and warm colors: reds and ochres (Figure 14b).

The concept of using polychrome in the Centennial Hall was closely related to the use of glass and stained glass windows. Glass, white and colored, was to shape the aesthetics of the new architecture in harmony with the raw concrete. Multicolored stained glass windows were to adorn the dome windows on all levels.

The motif of glass architecture appeared in Scheerbart's poems, in which glass—due to its transparency—was close to the structure of crystal, fascinating the artists of modernism. Crystal, as Peter Behrens put it, symbolized by its nature the transformation of "raw unformed life into an existence of order and beauty", it was "a symbol of new life"—new art. The later famous Crystal Palace, erected for the first world exhibition in London in 1851 to a design by Joseph Paxton, became an inspirational work. The exhibition pavilion made of iron and glass was for architects of Berg's generation a symbol of overcoming historical styles by dematerializing architecture and directing it towards "pure form" (Prange 1994). It embodied "the idea of absolute mastery over nature" and an unwavering faith in the potential of technology (Kohlmaier and von Sartory 1988, p. 33). His fascination with the structure of crystal also contributed to his interest in the "crystalline lines of the Gothic". As Bruno Taut wrote, "the Gothic cathedral is a prelude to the architecture of glass" (Wolter 1955, p. 657). He also argued that the space of the Gothic cathedral—with its characteristic light—leads people to God, and with the use of new technical possibilities, it can be transformed into a "cathedral of the future", where people will be able to experience the reflection of the cosmos in the spiral of colors of the glass dome (Wolter 1955, p. 568).

The prototypes of the crystalline form and light effects of glass architecture were found in Gothic religious buildings. Max Berg, as well as Hans Poelzig, who came from Carl Schäfer's neo-Gothic school, were familiar with these theories. Hence, they were also familiar with Paul Scheerbart's visions of architecture made of glass and light, which were published in the 1890s, long before *Gläserne Architektur* was published in 1914. Paul Scheerbart cooperated with the *Der Morgen* magazine. Its publisher was Werner Sombart, a member of the Breslau circle of intellectuals gathered around Albert and Toni Neisser and Carl Hauptmann (Scheerbart 1907). Max Berg also belonged to this circle. Paul Scheerbart's inclination towards mysticism might have aroused Berg's, Poelzig's, and Hauptmann's interest. It is possible that ideas connected with glass architecture were also discussed in this circle.

Almost simultaneously with the Centennial Hall, Hans Poelzig's famous Upper Silesian Tower, inspired by glass architecture, was built in Poznań. The architect was able to realize what Max Berg had to give up in Wroclaw, namely the use of colored glass in the core of the tower. The ideas of glass architecture that underpinned the construction of the Centennial Hall and the Upper Silesian Tower can be seen particularly clearly in the evocative night views of both buildings, which were distributed, among other things, in the form of postcards. The tower, during the day a visible sign of the East German Exhibition, was transformed at night into a kind of lighthouse, whose illumination was additionally

strengthened by a moving reflector mounted on the dome's canopy. In this way, Paul Scheerbart's vision of "spotlights on all glass towers" was realized (Störtkuhl 2000, pp. 380–81).

Both the Upper Silesian Tower and the Centennial Hall of Max Berg influenced the designs of two works by Bruno Taut, in which the architect realized the idea of "glass architecture"—in the so-called Iron Monument at the Leipzig Exhibition in 1913 and in the Glass Pavilion at the Werkbund Exhibition in Cologne in 1914. Illuminated from within, the pavilion resembled with its glow a gigantic lantern erected from glass and iron (Speidel 1995).

The glazed Centennial Hall evokes similar associations, and if it had been decorated according to the architect's original intentions, it too would have shimmered in various colors like Taut's pavilions. This is also indicated by the shape of the lantern, whose construction resembles the Bruno Taut pavilion from the Cologne exhibition. It should be emphasized that unlike the works of Poelzig and Taut, the Centennial Hall implemented the idea of "glass architecture" in reinforced concrete construction and raw concrete. Unfortunately, the described plans of architect Max Berg and artist Oskar Kokoschka never came to fruition.

### 3.3. Building Research

Stratigraphic studies of the paintings and archival searches reveal a new face of this building. The research was conducted under the CMP program (Conservation Management Plan—prepared by the authors of this article, among others)—a grant from The Getty Foundation under the "Keeping It Modern" program. According to the source materials, the architect intended to leave only the exterior part raw, while the interior was to be decorated with paintings, stained glass, and sculpture. Stratigraphic research has shown that the exterior walls were covered with a transparent yellowish color coating (Figure 15).

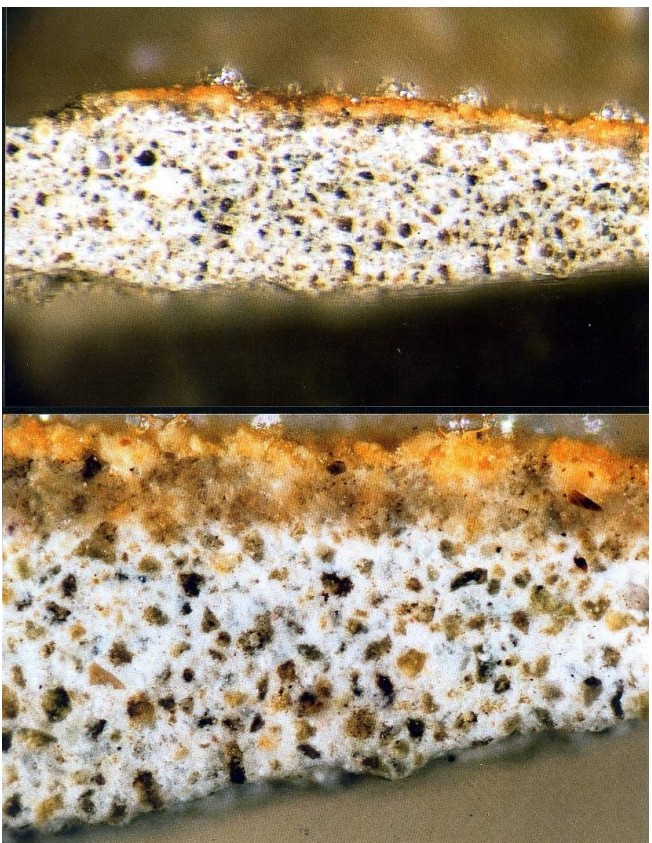

**Figure 15.** Centennial Hall before renovation in 2009; stratigraphies showing a yellowish outer layer (photo R. Wójtowicz).

Centennial Hall has undergone many renovations in the post-war period, but it was not until 2009 that the process of the proper restoration of this concrete building began. Traces of wooden formwork were visible on the outer surface of the concrete. In 1913, the architect chose a specific, unprecedented aesthetic of raw concrete. In many areas not exposed to the weather, the yellowish remains of the exterior coating applied to the entire concrete surface were visible. The original color of the façade on the underside of the roof of the connecting building between the hall and the restaurant has been remarkably well preserved.

Stratigraphic studies by Ryszard Wójtowicz (2016) confirmed the application of a very thin layer with yellowish mineral pigment. It was a thin mineral coating, lime paint with iron pigment, fixed and additionally protected with water glass. The only unknown was the time when such a paint layer was made. Certainly, it was done by order of the designer Max Berg. Probably, the differences between the batches of the building performed on consecutive days caused such a decision.

The color of the concrete was to be complemented by the color of the wooden window woodwork and the color of the panes. The original woodwork was made of hardwood, a variety of mahogany, the so-called ironwood with a rusty red color. The windows used ornamental glass in a yellowish color, and the lantern crowning the cupola was glazed in red (Figure 16).

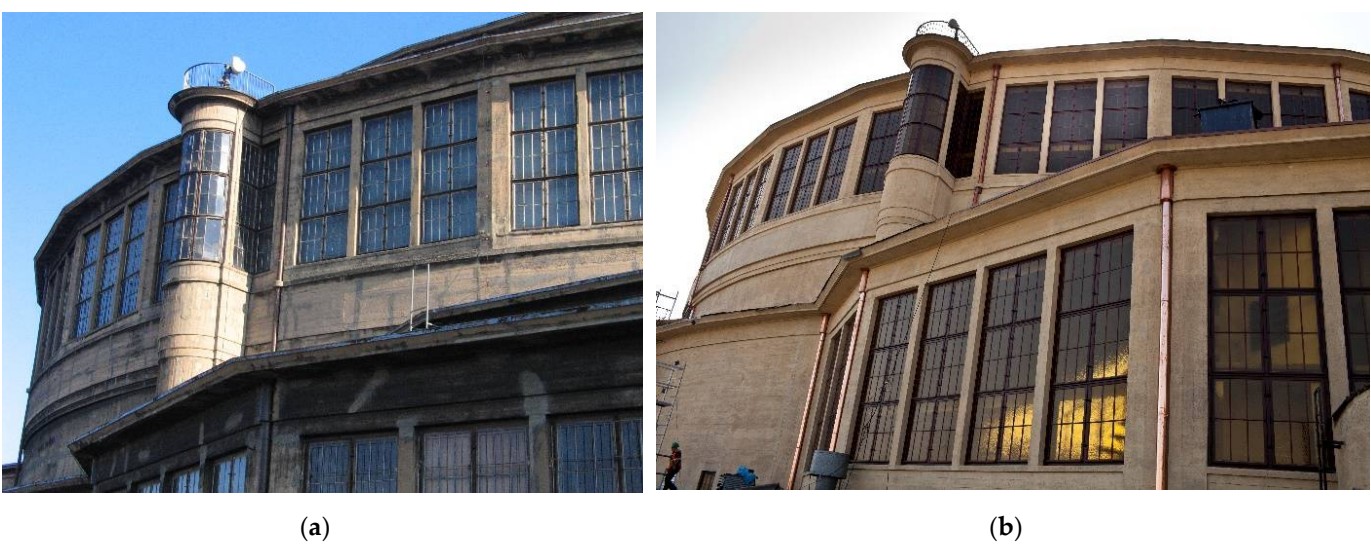

(**a**)　　　　　　　　　　　　　　　　　　　　　(**b**)

**Figure 16.** Centennial Hall: (**a**) before renovation (photo J. Ilkosz) and (**b**) after renovation in 2009 (photo J. Ilkosz).

During the restoration work, it was decided to restore the yellowish semi-transparent coating with which the concrete was originally covered and the yellow cathedral glass (according to the original documentation found and the former manufacturer's pattern) (Figure 17).

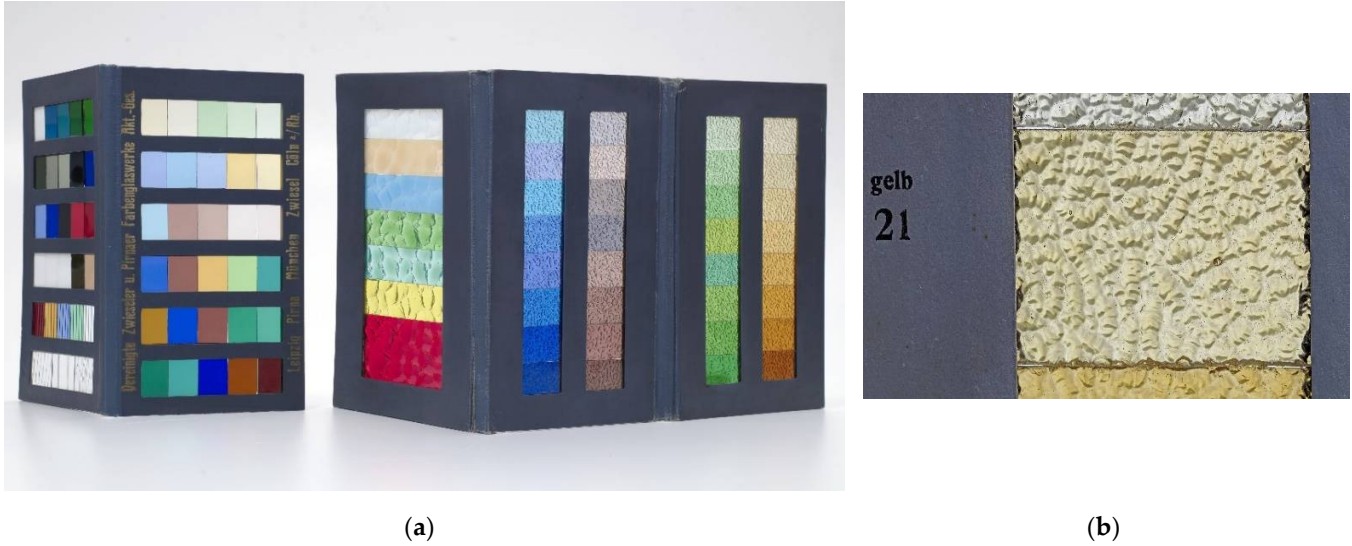

(**a**)

(**b**)

**Figure 17.** Template of the former glass manufacturer: (**a**) a whole range of colors and (**b**) ornamental glass in a yellowish color used during the renovation of Centennial Hall, in accordance with the original documentation.

## 4. Conclusions

The Centennial Hall is a work of architecture that arouses interest not only because of its unique structure, material, and innovative form but also because of what was not realized, which was, for the time, a daring artistic project. The value of the hall lies not only in its material form, but also in what has survived only in the form of drawings, sketches, and descriptions. Hiding there, Berg's extremely innovative idea to decorate the building deserves its rightful place in history.

Centennial Hall should be understood as a kind of *Gesamtkunswerk*, as intended by the author. It is a combination of innovative construction, form, and color palette. It seems to be an exception in comparison to other reinforced concrete buildings in Germany and Europe of that period and confirms the thesis that each building is an individual statement of an architect, an artist who wants to stand out and not follow the paths indicated by others.

Proof of the building's unique character is its inclusion on the UNESCO World Heritage List in 2006.

The condition of the Centennial Hall, now more than 100 years old, is very good. The methods of recent restoration of the building guarantee its survival for centuries to come. As part of the development of the CMP (Conservation Management Plan) in 2015, further studies were carried out to confirm the correctness of the choice of conservation methods.

The 20th century was a century of extraordinary changes in architecture, a century that saw a revolution in architectural form. Certainly, Max Berg was a co-creator of this revolution and with the aesthetics of his outstanding project in Wrocław—the Centennial Hall—he changed the architecture of the 20th century for good.

**Author Contributions:** Conceptualization, J.I., R.W. and J.U.; methodology, J.I., R.W. and J.U.; software, J.I.; validation, J.I., R.W. and J.U.; formal analysis, J.I. and J.U.; investigation, J.I., R.W. and J.U.; writing—original draft preparation, J.I., R.W. and J.U.; writing—review and editing, J.I., R.W. and J.U.; visualization, J.I., R.W. and J.U.; supervision, J.U.; project administration, J.I.; funding acquisition, J.I., R.W. and J.U. All authors have read and agreed to the published version of the manuscript.

**Funding:** This research received funding in September, 2014 from The Getty Foundation as part of the program *Keeping it modern* The Getty Foundation philanthropic initiative focused on conserving 20th century architecture—Centennial Hall was among the first ten grants awarded to conserve iconic modern architecture around the globe.

**Institutional Review Board Statement:** Not applicable.

**Informed Consent Statement:** Not applicable.

**Data Availability Statement:** The study did not report any data.

**Conflicts of Interest:** The authors declare no conflict of interest.

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
