# Peer review of "New Form, New Material and Color Scheme, the Exposed Concrete Phenomenon—The Centennial Hall in Wrocław"

_arts, 1910_

Round 1
Reviewer 1 Report
The author writes in his Abstract the location of Breslau (now Wroclaw PL) as "eastern Germany". This is not very lucky chosen, because there does not exist a country in a political sense "Eastern Germany", but rather a Deutsches Kaiserreich, German Reich or German Empire in those days. In a cultural sense one spoke of Ostdeutsche Ausstellung but in General I would prefer one of the three named descriptions.
Positiv to mention is that the cultural importance and historiography on the Centenial Hall has been well and thoroughly expressed by the author`s long introduction text, especially in archtectural style (citing historic parallel buildings), and construction level.
On the other hand, the article is completely focussing on paint (external yellow appearance) and collor (yellow stained glass, and other glass planned) as well in a lesser degree on interior painting. Why the political symbology in 1910 is not discussed.? What is the impact of the used colors. More in extra file. Please note error line 123 likened sould be linked.

Reviewer 2 Report
the reviewer does not raise any comments
Reviewer 3 Report
The article aims to fill an important gap in architectural research. It makes a very significant contribution to research on the topic of 20th-century architecture, especially on the combination of multicolored decorations with rawness of concrete. It is original and novel. The issues discussed in the article are important from the scientific point of view. The objectives of the article are formulated very clearly. The argumentation is logical, the article has the correct structure, and is written carefully. The article refers to existing research results, the cited literature is adequate and sufficient with regard to the discussed issues.
